# Study on Chemical Constituents of *Panax notoginseng* Leaves

**DOI:** 10.3390/molecules28052194

**Published:** 2023-02-27

**Authors:** Xiaojuan Sun, Hongbo Deng, Tengyun Shu, Min Xu, Lihua Su, Haizhou Li

**Affiliations:** Center for Pharmaceutical Sciences, Faculty of Life Science and Technology, Kunming University of Science and Technology, 727 Jingming South Road, Chenggong District, Kunming 650500, China

**Keywords:** *Panax notoginseng* leaves, protopanaxadiol saponins, SH-SY5Y cells, nerve injury

## Abstract

*Panax notoginseng* (Burk.) F. H. is a genuine medicinal material in Yunnan Province. As accessories, *P. notoginseng* leaves mainly contain protopanaxadiol saponins. The preliminary findings have indicated that *P. notoginseng* leaves contribute to its significant pharmacological effects and have been administrated to tranquilize and treat cancer and nerve injury. Saponins from *P. notoginseng* leaves were isolated and purified by different chromatographic methods, and the structures of **1**–**22** were elucidated mainly through comprehensive analyses of spectroscopic data. Moreover, the SH-SY5Y cells protection bioactivities of all isolated compounds were tested by establishing L-glutamate models for nerve cell injury. As a result, twenty-two saponins, including eight dammarane saponins, namely notoginsenosides SL_1_-SL_8_ (**1**–**8**), were identified as new compounds, together with fourteen known compounds, namely notoginsenoside NL-A_3_ (**9**), ginsenoside Rc (**10**), gypenoside IX (**11**), gypenoside XVII (**12**), notoginsenoside Fc (**13**), quinquenoside L_3_ (**14**), notoginsenoside NL-B_1_ (**15**), notoginsenoside NL-C_2_ (**16**), notoginsenoside NL-H_2_ (**17**), notoginsenoside NL-H_1_ (**18**), vina-ginsenoside R_13_ (**19**), ginsenoside II (**20**), majoroside F_4_ (**21**), and notoginsenoside LK_4_ (**22**). Among them, notoginsenoside SL_1_ (**1**), notoginsenoside SL_3_ (**3**), notoginsenoside NL-A_3_ (**9**), and ginsenoside Rc (**10**) showed slight protective effects against L-glutamate-induced nerve cell injury (30 µM).

## 1. Introduction

*Panax notoginseng* (Burk.) F. H. Chen, also called “Sanqi” in Chinese, which belongs to the *Panax* genus, family Araliaceae [1]. *P. notoginseng* is a valuable traditional Chinese medical herb, which has also been used as medicines for a long time, such as Yun-Nan-Bai-Yao, Xuesaitong capsules, and Xuesaitong injections [2]. As a genuine medicinal material in Yunnan, *P. notoginseng* is cultivated extensively in Wenshan on account of its unique geological and climatic conditions. Its roots have been widely used as tonic and main components in a great deal of compound preparations of Chinese medicine. *P. notoginseng* is one of the most widely used Chinese herbal drugs for the treatment of cardiovascular diseases, such as occlusive vasculitis, coronary diseases, atherosclerosis, and cerebral infarction in China and other overseas countries [3]. As resource accessories, few studies have been published on the leaves of *P. notoginseng* in the early stage [4]. However, *P. notoginseng* leaves play vital roles in medicinal and edible value. Modern pharmacological studies reveal that *P. notoginseng* leaves have shown remarkable effects as promising tranquilization [5], antidepressant [6,7], antioxidant [8], and anticancer treatments [9,10,11], as well as have shown multiple benefits on the blood system, cardiovascular system [12], nervous system [13], and metabolic system. Chinese patent medicine “QiyeShenAnPian” takes *P. notoginseng* leaves as raw material, and it has remarkable effects on invigorating qi, tranquilization, stimulating blood circulation, relieving pain, etc. Approximately 3 Mt *P. notoginseng* stem-leaves are produced in China annually, while they are used for forage or discarded in the local environment [14]. Therefore, the studies of *P. notoginseng* leaves have become a hot topic nowadays.

New research has revealed that *P. notoginseng* leaves have been recognized as a rich source of dammarane triterpenoid saponins, flavonoids, polysaccharides, peptides, polyacetylenes, and fatty acids. Among them, saponins as major active ingredients are mainly protopanaxadiol saponins [15]. Cao et al. [12] identified 226 saponins from *P. notoginseng* leaves by online two-dimensional liquid chromatography. Li et al. [16] extracted *P. notoginseng* leaves with ethanol, then isolated and identified 16 monomer compounds from ethanol extract, including ginsenosides Rg_1_, Rd, Re, Rb_1_, Rb_3_, Rg_3_, Rh_2_, F_2_, Rc, notoginsenoside Fd, ginsenoside Mx, gypenoside Mx, and notoginsenoside R_1_. Ruan et al. [17] isolated and identified 11 saponins with anti-inflammatory activity from *P. notoginseng* leaves: notoginsenosides NL-E_1_-NL-E_4_, NL-F_1_, NL-F_2_, NL-G_2_, NL-H_1_-NL-H_3_, and all of those are 20 (*S*)-protopanaxadiol saponins. Yang et al. [18] isolated two new dammarane-type triterpenoids from the stems and leaves of *P. notoginseng*, namely notoginsenoside SY1 and notoginsenoside SY2.

In this paper, dammarane triterpenoid saponins from *P. notoginseng* leaves were isolated and identified. Moreover, the neuroprotective effect of saponins was tested in SH-SY5Y cells induced by L-glutamate.

## 2. Results and Discussion

The extract of *P. notoginseng* leaves was isolated by silica gel, Sephadex LH-20, ODS, and preparative high-performance liquid chromatography (pre-HPLC), and eight new dammarane-type triterpenoid saponins, notoginsenosides SL_1_-SL_8_ (**1**–**8**) were yielded (shown in Figure 1).

### 2.1. Structural Elucidation

Compound **1** (4.0 mg): a white amorphous powder. The molecular formula was assigned as C_47_H_82_O_18_ by positive HR-ESI-MS spectrum at *m/z* 957.5231 [M + Na]^+^ (calcd. For. C_47_H_82_O_18_Na: 957.5542). ${\left[\alpha \right]}_{D}^{23}$ + 7.00 (c 0.19, MeOH). The IR spectrum illustrated the presence of hydroxyl (3416 cm^−1^). In the ^1^H-NMR spectrum (shown in Table 1), eight methyl groups *δ*_H_: 0.90 (3H, s), 0.81 (3H, s), 1.61 (3H, s), 1.59 (3H, s), 1.53 (3H, s), 1.30 (3H, s), 0.99 (3H, s), and 1.00 (3H, s), were perceived in the high field. In addition, four characteristic signals *δ*_H_: 3.43 (1H, dd, *J* = 4.8, 10.8 Hz), 3.65 (1H, m), 1.34 (1H, m), and 1.46 (1H, m) were presented. The ^13^C-NMR spectrum indicated 47 carbon signals (shown in Table 1), which included *δ*_C_ as: 88.6, 18.2, 70.2 and 83.0. The above indicated compound **1** was a protopanaxadiol saponin substituted by sugars at C-3 and C-20. A series of carbon signals (*δ*_C_: 70.5, 39.8, 23.6, 27.5, 26.4, 25.2, 24.8) and hydrogen signals (*δ*_H_: 1.93, 2.45, 1.81, 2.05) showed that compound **1** is a saponin without a double bond at C-24/25, and *δ*_C_: 70.5 of which was protons bearing an oxygenated quaternary carbon at C-25. The HMBC correlations from H_2_-23/H_3_-26/H_3_-27 to C-24, and H_3_-26/H_3_-27 to C-25 (shown in Appendix A), indicated that a hydroxyl was located at C-25. The ^1^H-NMR and ^13^C-NMR data of the side chain was similar to 20(*R*)-Dammarane-3*β*, 12*β*, 20, 25-tetrol [19], who displays no double carbon signal, and it showed a hydroxyl in this constituent (*δ*_C_: 70.5). 

Three anomeric carbon resonances (*δ*_C_: 106.8 (Glc H-1′), 97.9 (Glc C-1″), 105.5 (Xyl C-1‴)) of sugars were observed in the ^13^C-NMR spectrum. Besides, three hydrogen signals (*δ*_H_: 4.97 (d, *J* = 7.8 Hz, Glc H-1′), 5.12 (d, *J* = 7.8 Hz, Glc H-1″), 4.97 (d, *J* = 7.2 Hz, Xyl H-1‴)) of anomeric carbons were present from the ^1^H-NMR spectrum, and all were *β* configurations. The HMBC spectrum showed the correlations from Glc H-1′ (*δ*_H_: 4.97) to C-3 (*δ*_C_: 88.6), Glc H-1″ (*δ*_H_: 5.12) to C-20 (*δ*_C_: 83.0), and Xyl H-1‴ (*δ*_H_: 4.97) to C-6″ (*δ*_C_: 69.4) (shown in Figure 2), respectively, from which indicated Glc C-1′ was connected with C-3, and Glc C-6″ was connected with Xyl C-1‴, finally Glc C-1″ was connected with C-20. Multiple methods were applied to determine the configurations of sugars, such as hydrolysis, derivatization, and GC analysis. The ^1^H-NMR and ^13^C-NMR data of sugars was highly consistent with gypenoside IX [20], and the structure of this compound was then determined and named as notoginsenoside SL_1_.

Compound **2** (24.0 mg) was obtained as a white amorphous powder. Its molecular formula was determined as C_47_H_80_O_18_, evidenced by positive HR-ESI-MS data *m/z* 955.5245 [M + Na]^+^ (calcd. For. C_47_H_80_O_18_Na, 955.5232). ${\left[\alpha \right]}_{D}^{23}$ − 1.46 (c 0.20, MeOH). As shown in its IR spectrum, the signals of hydroxyl (3406 cm^−1^) and double bond (1230 cm^−1^) were significant. In the ^1^H-NMR spectrum (shown in Table 2), seven methyl groups (*δ*_H_: 0.93, 0.79, 1.63, 1.91, 1.31, 0.96 and 0.94, each 3H, s) were revealed. Moreover, signals [*δ*_H_: 3.37 (1H, dd, *J* = 4.8, 10.8 Hz), 3.12 (1H, m), 1.36 (1H, m), 1.47 (1H, m)] were presented in the ^1^H-NMR spectrum. In the ^13^C-NMR spectrum (shown in Table 2), 47 carbon signals were revealed, including *δ*_C_: 88.6, 18.2, 70.1, and 83.2. Compound **2** was a protopanaxadiol saponin substituted by sugars at C-3 and C-20 indicating from above. A set of carbon signals (*δ*_C_: 150.0, 110.3, 22.3, 32.6, 76.6, 30.6, 17.8) and hydrogen signals (*δ*_H_: 4.95, 5.26, 2.54, 2.24, 4.47, 4.95, 5.26, 1.91) indicated a hydroxyl was connected with C-24, and this compound was a saponin with a changeable side chain. Furthermore, the HMBC correlations from H_3_-27/H_2_-26/H_1_-24 to C-25, and H_2_-22/H_2_-26/H_3_-27 to C-24 showed the existence of a double bond between C-25 and C-26, and a hydroxyl at C-24 (shown in Appendix A). The absolute configuration of the hydroxyl at C-24 was determined by its chemical shift of ^13^C-NMR. The chemical shift of a relatively low field corresponded to the *R* configuration of C-24, and a relatively high field corresponded to the *S* configuration of C-24. As shown in the ^13^C-NMR spectrum, a hydroxyl existed in the relatively low field (*δ*_C_: 76.6), and indicated the configuration of C-24 was *R*. The ^1^H-NMR and ^13^C-NMR data of side chain was similar to majoroside F_1_ [21].

Three anomeric carbon signals (*δ*_C_: 106.8 (Glc C-1′), 97.9 (Glc C-1″), 109.8 (Ara(f) C-1‴) and hydrogen signals [*δ*_H_: 4.97 (d, *J* = 7.8 Hz, Glc H-1′), 5.17 (d, *J* = 7.7 Hz, Glc H-1″), 4.69 (br.s, Ara (f) H-1‴] were observed, and it was revealed that the configurations of two glucoses were *β*, and that arabinose (f) was *α*. In the HMBC spectrum, the correlations from Glc H-1′ (*δ*_H_: 4.97) to C-3 (*δ*_C_: 88.6), Glc H-1″ (*δ*_H_: 5.17) to C-20 (*δ*_C_: 83.2), and Ara (f) H-1‴ (*δ*_H_: 4.69) to C-6″ (*δ*_C_: 68.3), respectively, indicated that Glc C-1′ was connected with C-3, and Glc C-6″ was connected with Ara (f) C-1‴, and finally, Glc C-1″ was connected with C-20 (shown in Figure 2). Three glycosides were *β*-D-glucoses and *α*-L-arabinose determined by hydrolysis, derivatization, and GC analysis. The ^1^H-NMR and ^13^C-NMR data of sugars was highly consistent with notoginsenoside Fe [22]. Consequently, the structure of compound **2** was determined and named as notoginsenoside SL_2_.

Compound **3** (40.5 mg): white amorphous powder. The molecular formula of 3 was deduced to be C_47_H_80_O_19_ by positive HR-ESI-MS data at *m*/*z* 971.5255 [M + Na]^+^ (calculated for C_47_H_80_O_19_Na, 971.5242). ${\left[\alpha \right]}_{D}^{23}$ + 7.89 (c 0.18, MeOH). The IR absorptions revealed the existence of hydroxyl (3421 cm^−1^) and double bond (1261 cm^−1^). In the ^1^H-NMR spectrum (shown in Table 3), seven angular methyl groups (*δ*_H_: 0.97, 0.81, 1.63, 1.96, 1.32, 1.01 and 0.98, each 3H, s), and two hydrogen signals of oxymethine [*δ*_H_: 3.38 (1H, dd, J = 4.2 Hz, 11.6 Hz), 4.18 (1H, m)] were illustrated. The ^13^C-NMR spectrum indicated 47 carbon signals (shown in Table 3), including three characteristic carbon signals (*δ*_C_: 88.6, 18.2, and 83.0). Compound **3** was a protopanaxadiol saponin substituted by sugars at C-3 and C-20 from above. The ^1^H-NMR signals [*δ*_H_: 1.97, 2.23 (m, H_2_-23), 4.80 (m, H-24), 5.09, 5.27 (br.s, H_2_-26)] and ^13^C-NMR signals (*δ*_C_: 145.9, 113.3, 22.2, 23.6, 89.8, 26.0, 17.3) indicated that C-24 of this compound was substituted, and its lateral chain was changed. Combining the ^13^C-NMR (*δ*_C_: 89.8) with the data of MS, it revealed that C-24 of this compound was replaced by hydroxyperoxy. Besides, the HMBC correlations from H_2_-26/H_3_-27 to C-25, and H_2_-23/H_2_-26/H_3_-27 to C-24 verified that an alkene proton signal existed between C-25 and C-26 (shown in Appendix A). Furthermore, a hydroxyperoxy existed at C-24. As for the configuration of hydroxyperoxy at C-24, it was necessary to convert hydroxyperoxy into hydroxyl. The ^1^H-NMR and ^13^C-NMR data of side chain was similar to ginsenoside II [23]. 

Anomeric carbon signals [*δ*_C_: 106.8 (Glc C-1′), 97.9 (Glc C-1′′), 105.5(Xyl C-1′″)] and hydrogen signals [*δ*_H_: 4.97 (d, *J* = 7.6 Hz, Glc H-1′), 5.12 (br.s, Glc H-1″), 4.97 (d, *J* = 7.2 Hz, Xyl H-1‴)] were observed and revealed the configurations of two glucoses were *β*. In the HMBC spectrum, Glc H-1′ (*δ*_H_: 4.97) was correlated with C-3 (*δ*_C_: 88.6), Glc H-1″ (*δ*_H_: 5.12) was correlated with C-20 (*δ*_C_ 83.0), and Xyl H-1‴ (*δ*_H_: 4.97) was correlated with C-6″ (*δ*_C_: 68.3) (shown in Figure 2), respectively, from which it was indicated that Glc C-1′ was connected with C-3, and Glc C-6″ was connected with Xyl C-1‴, finally Glc C-1″ was connected with C-20. Three glycosides were *β*-D-glucoses and *β*-D-xylose, determined by hydrolysis, derivatization, and GC analysis. The ^1^H-NMR and ^13^C-NMR data of sugars was highly consistent with gypenoside IX [20]. Consequently, the structure of compound **3** was determined and named as notoginsenoside SL_3_.

Compound **4** (3.0 mg) was obtained as a white amorphous powder. Its molecular formula was determined as C_47_H_80_O_18_, evidenced by positive HR-ESI-MS data (*m*/*z* 955.5247 [M + Na]^+^, calculated for C_47_H_80_O_18_Na, 955.5641). ${\left[\alpha \right]}_{D}^{23}$ + 19.20 (c 0.15, MeOH). The IR absorptions revealed the existence of hydroxyl (3433 cm^−1^) and double bond (1260 cm^−1^). The ^1^H-NMR showed eight angular methyl groups (*δ*_H_: 0.87, 0.95, 1.48, 1.59, 1.53, 1.30, 0.99, and 1.02, each 3H, s) and four characteristic hydrogen signals [3.43 (1H, dd, *J* = 4.8 Hz, 10.8 Hz), 3.65 (1H, m), 1.34, 1.47 (2H, m)] (shown in Table 4). In its ^13^C-NMR spectrum, 47 carbon signals were indicated, including three characteristic carbon signals (*δ*_C_: 89.2, 18.5, 83.8) (shown in Table 4). Compound **4** was a protopanaxadiol saponin substituted by sugars at C-3 and C-20 indicating from above. The ^1^H-NMR signals [*δ*_H_: 6.09 (d, *J* = 15.5 Hz, H_1_-24), 6.23 (ddd, *J* = 5.8, 8.5, 15.5 Hz, H_1_-23) and ^13^C-NMR signals (*δ*_C_: 122.7, 142.2, 70.0, 17.7, 17.8) indicated that C-25 of this compound was substituted, and its lateral chain was changed. Furthermore, its ^13^C-NMR indicated a quaternary carbon replaced by hydroxyl at C-25. The HMBC correlations from H_1_-23/H_1_-24/H_3_-26/H_3_-27 to C-25 and H_2_-22/H_1_-23/H_3_-26/H_3_-27 to C-24 indicated that an alkene proton signal existed between C-23 and C-24 (shown in Appendix A), and a hydroxyl at C-25. The ^1^H-NMR and ^13^C-NMR data of the side chain was similar to quinquenoside L3 [24].

Anomeric carbon signals [*δ*_C_: 106.8 (Glc C-1′), 97.9 (Glc C-1″), 104.6 (Ara (p) C-1‴)] and hydrogen signals [*δ*_H_: 4.93 (d, *J* = 7.8 Hz, Glc H-1′), 5.14 (d, *J* = 7.8 Hz, Glc H-1″), 5.00 (d, *J* = 6.0 Hz, Ara (p) C-1‴)] were observed, and revealed that the configurations of two glucoses were *β* and the configuration of an arabinose was *α*. Besides, the carbon signals of sugars [104.6 (Ara (p) C-1‴), 72.1 (Ara (p) C-2‴), 74.1 (Ara (p) C-3‴), 68.5 (Ara (p) C-4‴), and 65.6 (Ara (p) C-5‴)] revealed that this arabinose was a pyranose. In the HMBC spectrum, Glc H-1′ (*δ*_H_: 4.93) was correlated with C-3 (*δ*_C_: 89.2). Glc H-1″ (*δ*_H_: 5.14) was correlated with C-20 (*δ*_C_: 83.8), and Ara (p) H-1‴ (*δ*_H_: 5.00) was correlated with Glc C-6″ (*δ*_C_: 68.3) (shown in Figure 2), respectively. It was indicated Glc C-1′ was connected with C-3, and Glc C-6″ was connected with Ara (p) C-1‴, and finally, Glc C-1″ was connected with C-20. The configurations of glycosides were *β*-D-glucoses and *α*-L- arabinose determined by similar methods above. The ^1^H-NMR and ^13^C- NMR data of sugars was highly consistent with ginsenoside Rd_2_. Then, the structure of compound 4 was elucidated and named as notoginsenoside SL_4_.

Compound **5** (57.1 mg) was obtained as a white amorphous powder. The molecular formula was deduced to be C_58_H_98_O_28_ by positive HR-ESI-MS data at *m*/*z* 1265.6144 [M + Na]^+^ (calculated for C_58_H_98_O_28_Na, 1265.6142). ${\left[\alpha \right]}_{D}^{23}$ + 10.67 (c 0.15, MeOH). The existence of hydroxyl (3417 cm^−1^) and double bond (1258 cm^−1^) was revealed from its IR spectrum. In its ^1^H-NMR spectrum (shown in Table 5), seven angular methyl groups (*δ*_H_: 0.96, 0.80, 1.64, 1.96, 1.28, 1.11 and 0.95, each 3H, s), and four characteristic hydrogen signals [*δ*_H_: 3.30 (dd, *J* = 3.9, 11.5 Hz, 1H), 1.34 (m, 1H), 1.47 (m, 1H), 4.18 (m, 1H)] were indicated. The ^13^C-NMR spectrum revealed 58 carbon signals (shown in Table 5), including four characteristic carbon signals (*δ*_C_: 88.6, 18.2, 69.9, and 83.2). Compound **5** was a protopanaxadiol saponin substituted by sugars at C-3 and C-20 from above. The ^1^H-NMR signals [*δ*_H_: 1.97, 2.20 (m, H_2_-23), 4.80 (t, *J* = 6.7 Hz, H-24), 5.09, 5.28 (br.s, H_2_-26)], and ^13^C-NMR signals (*δ*_C_: 146.1, 113.1, 32.7, 90.0, 26.4, 17.3) indicated that C-24 of this compound was substituted and a double bond was revealed between C-25 and C-26. Combining the ^13^C-NMR (*δ*_C_: 90.0) with the data of MS, it revealed that C-24 of this compound was replaced by hydroxyperoxy. Besides, the HMBC correlations from H_2_-23/H_2_-26/H_3_-27 to C-24, and H_2_-26/H_3_-27 to C-25 verified that an alkene proton signal existed between C-25 and C-26 (shown in Appendix A), and a hydroxyperoxy existed at C-24. The ^1^H-NMR and ^13^C-NMR data of the side chain was similar to ginsenoside Ⅱ [23].

Anomeric carbon signals [*δ*_C_: 104.6 (Glc C-1′), 103.0 (Glc C-1″), 106.2 (Xyl C-1‴), 97.8 (Glc H-1′′′′), 105.5 (Xyl C-1′′′′′)] and hydrogen signals [(*δ*_H_: 4.95 (d, *J* = 7.7 Hz, Glc H-1′), 5.54 (d, *J* = 6.8 Hz, Glc H-1″), 5.44 (d, *J* = 6.8 Hz, Xyl H-1‴), 5.12 (br.s, Glc H-1′′′′), and 5.01 (d, *J* = 7.4 Hz, Xyl H-1′′′′′)] were observed and revealed that the configurations of three glucoses were *β*. In the HMBC spectrum, Glc H-1′ (*δ*_H_: 4.95) was correlated with C-3 (*δ*_C_ 88.6), Glc H-1″ (*δ*_H_: 5.54) was correlated with Glc C-2′ (*δ*_C_: 82.3), Xyl H-1‴ (*δ*_H_: 5.44) was correlated with C-2″ (*δ*_C_: 84.2), Glc H-1′′′′ (*δ*_H_: 5.12) was correlated with C-20 (*δ*_C_: 83.2), and Xyl H-1′′′′′ (*δ*_H_ 5.01) was correlated with Glc C-6′′′′ (*δ*_C_: 69.9) (shown in Figure 2), respectively. From this it was indicated that Glc C-1′ was connected with C-3, Glc C-1″ was connected with Glc C-2′, Glc C-1‴was connected with Xyl C-2″, and Glc C-1′′′′ was connected with C-20, and finally Xyl C-1′′′′′ was connected with Glc C-6′′′′. The glycosides were *β*-D-glucoses and *β*-D-xyloses, which were determined by hydrolysis, derivatization, and GC analysis. The ^1^H-NMR and ^13^C-NMR data of sugars was highly consistent with notoginsenoside Fc [25]. Then, the structure of compound **5** was determined and named as notoginsenoside SL_5_.

Compound **6** (10.0 mg) was a white amorphous powder. The molecular formula of 6 was deduced to be C_58_H_98_O_28_ by positive HR-ESI-MS data at *m*/*z* 1265.6149 [M + Na]^+^ (calculated for C_58_H_98_O_28_Na, 1265.6142). ${\left[\alpha \right]}_{D}^{23}$ + 7.45 (c 0.15, MeOH). The IR absorptions revealed the existence of hydroxyl (3415 cm^−1^) and double bond (1258 cm^−1^). In the ^1^H-NMR spectrum (shown in Table 6), eight angular methyl groups (*δ*_H_: 1.01, 0.83, 1.63, 1.62, 1.62, 1.29, 1.12, and 0.92, each 3H, s) were shown. Four characteristic hydrogen signals [*δ*_H_: 3.30 (dd-like, 1H), 1.38 (m, 1H), 1.54 (m, 1H), 4.07 (m, 1H)] were revealed. The ^13^C-NMR spectrum indicated 58 carbon signals, including four characteristic carbon signals (*δ*_C_: 88.6, 18.2, 70.2, and 83.0) (shown in Table 6). Compound **6** was a protopanaxadiol saponin substituted by sugars at C-3 and C-20 from above. According to HSQC, H-23 (*δ*_H_: 6.20, m) was correlated with C-23 (*δ*_C_: 126.5), and H-24 (*δ*_H_: 6.15, br.s) was correlated with C-24 (*δ*_C_: 137.8), which indicated that a double bond existed between C-23 and C-24. Combining MS spectrum with carbon spectrum (*δ*_C_: 81.1), a hydroperoxyl was presented at C-25. Furthermore, the HMBC correlations from H_3_-26/H_3_-27/H_1_-24/H_1_-23 to C-25 and H_3_-27/H_3_-26/H_1_-23/H_2_-22 to C-24 verified that an alkene proton signal existed between C-23 and C-24, and a hydroxyperoxy existed at C-25 (shown in Appendix A). The ^1^H-NMR and ^13^C-NMR data of the side chain was similar to notoginsenoside E [26].

Anomeric carbon signals [*δ*_C_: 104.6 (Glc C-1′), 102.9 (Glc C-1″), 106.2 (Xyl C-1‴), 98.0 (Glc C-1′′′′), 105.4 (Xyl C-1′′′′′)] and hydrogen signals [*δ*_H_: 4.96 (d, *J* = 6.2 Hz, Glc H-1′), 5.54 (d-like, Glc H-1″), 5.45 (d, *J* = 6.0 Hz, Xyl H-1‴), 5.20 (br.s, Glc H-1′′′′), 5.00 (d, *J* = 6.8 Hz, Xyl H-1′′′′′)] were observed, and the configurations of five glycosyl signals were all *β*. In the HMBC spectrum, the correlations from Glc H-1′ (*δ*_H_: 4.96) to C-3 (*δ*_C_: 88.6), Glc H-1″ (*δ*_H_: 5.54) to Glc C-2′ (*δ*_C_: 82.7), Xyl H-1‴ (*δ*_H_: 5.45) to Glc C-2″ (*δ*_H_: 84.3), Glc H-1′′′′ (*δ*_H_: 5.20) to C-20 (*δ*_C_: 83.0), and Xyl H-1′′′′ (*δ*_H_: 5.00) to Glc C-6′′′′ (*δ*_C_: 69.7) (shown in Figure 2), respectively, from which indicated Glc C-1′ was connected with C-3, Glc C-1″ was connected with Glc C-2′, Glc C-1‴ was connected with Xyl C-2″, and Glc C-1′′′′ was connected with C-20, finally Xyl C-1′′′′′ was connected with Glc C-6′′′′. Five glycosides were determined as *β*-D-glucoses and *β*-D-xyloses by same methods above. The ^1^H-NMR and ^13^C-NMR data of sugars was highly consistent with notoginsenoside Fc [27]. Finally, the structure of compound **6** was elucidated and named as notoginsenoside SL_6_.

Compound **7** (24.6 mg) was obtained as a white amorphous powder. The molecular formula of 7 was deduced to be C_58_H_98_O_28_ by positive HR-ESI-MS data at *m*/*z* 1265.6146 [M + Na]^+^ (calculated for C_58_H_98_O_28_Na, 1265.6142). ${\left[\alpha \right]}_{D}^{24}$ + 4.40 (c 0.18, MeOH). The IR absorptions revealed the existence of hydroxyl (3425 cm^−1^) and double bond (1257 cm^−1^). In its 1H-NMR spectrum (shown in Table 7), there were eight angular methyl groups (*δ*_H_: 0.79, 0.93, 1.63, 1.61, 1.61, 1.27, 1.14, and 0.94, each 3H, s). Four characteristic hydrogen signals [*δ*_H_: 3.30 (dd, J = 4.2, 11.4 Hz, 1H), 1.37 (m, 1H), 1.55 (m, 1H), 4.13 (m, 1H)] were revealed. The ^13^C-NMR spectrum indicated 58 carbon signals, including four characteristic carbon signals (*δ*_C_: 88.6, 18.2, 70.5 and 83.3) (shown in Table 7). Compound **7** was a protopanaxadiol saponin substituted by sugars at C-3 and C-20 from above. According to the hydrogen spectrum [*δ*_H_: 5.20 (m, H-23), 6.15 (br.s, H-24)] and carbon spectrum [*δ*_C_: 126.2 (C-23), 137.8 (C-24)], a double bond existed between C-23 and C-24. In combining MS spectrum with carbon spectrum (*δ*_C_: 81.1), a hydroxyperoxy was presented at C-25. Furthermore, the HMBC correlations from H_3_-26/H_3_-27/H_1_-24/H_1_-23 to C-25, and H_3_-27/H_3_-26/H_1_-23/H_2_-22 to C-24 verified that an alkene proton signal existed between C-23 and C-24 (shown in Appendix A), and a hydroxyperoxy existed at C-25. The ^1^H-NMR and ^13^C-NMR data of side chain was similar to notoginsenoside E [26].

Anomeric carbon signals (*δ*_C_: 104.6 (Glc C-1′), 102.9 (Glc C-1″), 106.2 (Xyl C-1‴), 98.0 (Glc C-1′′′′), 104.1 (Ara (p) C-1′′′′′) and hydrogen signals [*δ*_H_: 4.95 (d, *J* = 6.2 Hz, Glc H-1′), 5.52 (d, *J* = 7.8 Hz, Glc H-1″), 5.43 (d, *J* = 6.6 Hz, Xyl H-1‴), 5.20 (br.s, Glc H-1′′′′), 5.00 (d, *J* = 6.8 Hz, Ara (p) H-1′′′′′)] were observed, what were the signals of five glycosyl and the configurations of glucoses were *β* and arabinose was *α*. In the HMBC spectrum, Glc H-1′ (*δ*_H_: 4.95) was correlated with C-3 (*δ*_C_: 88.6). Glc H-1″ (*δ*_H_: 5.52 ) was correlated with Glc C-2′ (*δ*_C_: 82.7), Xyl H-1‴ (*δ*_H_: 5.43) was correlated with Glc C-2″ (*δ*_H_: 84.2), Glc H-1′′′′ (*δ*_H_: 5.20) was correlated with C-20 (*δ*_C_: 83.3), and Ara (p) H-1′′′′′ (*δ*_H_: 5.00) was correlated with Glc C-6′′′′ (*δ*_C_: 68.7) (shown in Figure 2), respectively, from which indicated Glc C-1′ was connected with C-3, Glc C-1″ was connected with Glc C-2′, Glc C-1‴ was connected with Xyl C-2″, and Glc C-1′′′′ was connected with C-20, and finally, Ara (p) C-1′′′′′ was connected with Glc C-6′′′′. Five glycosides were *β*-D-glucoses, *β*-D-xylose and *α*-L-arabinose, which was determined by same methods above. The ^1^H-NMR and ^13^C-NMR data of sugars was highly consistent with notoginsenoside Fz [4]. Accordingly, the structure of compound **7** was determined and named as notoginsenoside SL_7_.

Compound **8** (6.9 mg) was obtained as a white amorphous powder. The molecular formula of 8 was deduced to be C_58_H_96_O_26_ by positive HR-ESI-MS data at *m*/*z* 1232.4306 [M + Na]^+^ (calculated for C58H96O26Na, 1232.3532). ${\left[\alpha \right]}_{D}^{24}$ − 7.20 (*c* 0.32, MeOH). The IR absorptions revealed the existence of hydroxyl (3412 cm^−1^) and double bond (1265 cm^−1^). In its 1H-NMR spectrum (shown in Table 8), eight angular methyl groups (*δ*_H_: 0.88, 0.92, 1.61, 1.65, 1.94, 1.27, 1.10, 0.79, each 3H, s) were shown, including two characteristic methyl groups linked to a sp2 bond (*δ*_H_: 1.65 and 1.94). The 1H-NMR of 8 showed only an olefinic proton at *δ*_H_ 6.02 (d, J = 7.6 Hz, 1H). On the basis of four characteristic carbon signals (*δ*_C_: 89.1, 18.5, 79.1 and 83.6), compound 8 was a protopanaxadiol saponin substituted by sugars at C-3 and C-20 from above. Furthermore, the HMBC correlations from H_2_-22/H_1_-23/H_3_-26/H_3_-27 to C-24 and H_1_-25/H_3_-26/H_3_-27 to C-25 verified that an alkene proton signal existed between C-24 and C-25 (shown in Appendix A). In addition, the HMBC correlations from H_1_-12/H_2_-22/H_1_-24 to C-23 and the positive HR-ESI-MS data showed a molecule oxygen between C-12 and C-23. The ^1^H-NMR and ^13^C-NMR data of the side chain was similar to quinquefoloside-Lb [28], whose ^1^H-NMR and ^13^C-NMR data was assigned by comparing it with that in the literature.

Anomeric carbon signals (*δ*_C_: 105.3 (Glc C-1′), 105.6 (Glc C-1″), 106.2 (Xyl C-1‴), 98.4 (Glc C-1′′′′), 110.3 (Ara (f) C-1′′′′′) and hydrogen signals [ *δ*_H_: 4.96 (d, *J* = 8.3 Hz, Glc H-1′), 5.40 (d, *J* = 6.9 Hz, Glc H-1″), 5.44 (d, *J* = 6.6 Hz, Xyl H-1‴), 5.18 (br.s, Glc H-1′′′′), 5.68 (d, *J* = 6.8 Hz, Ara (f) H -1′′′′′) ] were observed, which revealed five glycosyl signals existed and the configurations of glucoses were *β* and arabinose was *α*. In the HMBC spectrum, Glc H-1′ (*δ*_H_: 4.96) was correlated with C-3 (*δ*_C_: 89.1). Glc H-1″ (*δ*_H_: 5.40) was correlated with Glc C-2′ (*δ*_C_: 83.2), Xyl H-1‴ (*δ*_H_: 5.44) was correlated with Glc C-2″ (*δ*_H_: 75.2), Glc H-1′′′′ (*δ*_H_: 5.18) was correlated with C-20 (*δ*_C_: 83.6), and Ara (f) H-1′′′′′ (*δ*_H_: 5.68) was correlated with Glc C-6′′′′ (*δ*_C_: 67.2) (shown in Figure 2), respectively, from which it was indicated that Glc C-1′ was connected with C-3, Glc C-1″ was connected with Glc C-2′, Glc C-1‴ was connected with Xyl C-2″, and Glc C-1′′′′ was connected with C-20, finally Ara (f) C-1′′′′′ was connected with Glc C-6′′′′. Five glycosides were *β*-D-glucoses, *β*-D-xylose and *α*-L- arabinose determining by same methods above. The ^1^H-NMR and ^13^C-NMR data of sugars was highly consistent with notoginsenoside NL-A_3_ [29]. Consequently, the structure of compound **8** was elucidated and named as notoginsenoside SL_8_.

### 2.2. Bioactivity Assays

In order to clarify the neuroprotective effect of saponins from *P. notoginseng* leaves, all the isolates were tested on L-glutamate-induced cellular damage in SH-SY5Y neuroblastoma cells by using MTT assays, and VPA was used as a positive control. The concentrations of L-glutamate and VPA were determined with gradient screening method. As a result, N-SL_1_ (**1**), N-SL_3_ (**3**), N-NL-A_2_ (**9**), and G-Rc (**10**) displayed slight activities at 30 µM (Figure 3). Then, under this concentration, in vitro potential neuroprotective activities of those compounds were investigated.

### 2.3. Discussion

Studies have reported that protopanaxadiol saponins are the main active ingredient in *P. notoginseng* leaves. In recent years, more and more rare saponins with changed side chains in *P. notoginseng* leaves have been continuously discovered, and most of the rare saponins retain intact sugar chains, mainly characterized by their side chains.

As shown in results, 22 triterpene saponins from *P. notoginseng* leaves were isolated and identified. Eight saponins were identified as new compounds, and they all were saponins with changed side chains. Among them, compound **1**, **2**, **4** featured 25-hydroxyl, 25-ene-24-hydroxyl, and 23-ene-25-hydroxyl. Compounds **3** and **5** featured 25-ene-24 hydroxyperoxy. Compounds **6**–**7** featured 23-ene-25 hydroxyperoxy. Additionally, compound **8** featured an oxygen ring. These eight new compounds are all protopanaxadiol saponins, which further verifies the theory that protopanaxadiol saponins are mainly contained in *P. notoginseng* leaves. The rare variant ginsenosides in *P. notoginseng* leaves are more abundant than those in *P. notoginseng* roots. The growth environment of *P. notoginseng* leaves and *P. notoginseng* roots is different, which may be the main reason why *P. notoginseng* leaves are more abundant in rare variant ginsenosides. The Yunnan area, with a high-altitude and strong sunshine climate, provides strong ultraviolet environment, and the chlorophyll is used as a photosensitizer. Therefore, it is speculated that the main component saponins of *P. notoginseng* leaves may be oxidized due to strong ultraviolet environmental factors, and the structures of the side chains are changed.

The saponins in *P. notoginseng* have neuroprotective effects. Because the main active components of *P. notoginseng* leaves are also saponins, it is speculated that the saponins in *P. notoginseng* leaves also have neuroprotective activity. In this study, the neuroprotective activity of *P*. *notoginseng* leaves was described, and the study found that the monomeric compounds **1**, **3**, **9,** and **10** had neuroprotective activity (30 µM). Indeed, the investigation of *P. notoginseng* leaves will provide valuable information in understanding the chemical constituents of *P. notoginseng* leaves and searching new candidates for neuroprotection agents. The studies of the plant itself and the isolates are now in progress, which may provide the basic theory for following research.

## 3. Materials and Methods

### 3.1. Plant Materials

*P. notoginseng* plants were collected from a market. The voucher specimen was identified by associate Prof. Haizhou Li and Prof. Min Xu, and was kept in the Department of Pharmaceutical Chemistry and Biology’s labratory at Kunming University of Science and Technology (Kunming, China). Yunnan Weihe Pharmaceutical Co., Ltd. (Yuxi, China) was entrusted to extract a sample with 60% ethanol, and made desugaring treatment with microporous adsorption resin which was then refined to get the total saponin of *P. notoginseng* leaves (Total yield was about 4%).

### 3.2. Experimental Instruments

The thin layer analysis was completed by three function ultraviolet analyzers (ZW-3, Jinan Sanquan Zhongshi Experimental Instrument Co., Ltd., Jinan, China). Respectively, optical rotations spectra were measured by JASCO DIP-370 (JASCO Co., Tokyo, Japan). The UV rotations spectra was measured by Shimadzu UV 2401 PC (Shimadzu, Japan). NMR analyses were carried out on a Bruker 600 MHz spectrometer (Bruker-Biospin group, Fällanden, Switzerland). HR-ESI-MS data was recorded on Waters AutoSpec Premier P776 (Agilent Technologies, Santa Clara, CA, USA). The HPLC-UV analyses were performed on a Waters 2695 series system (Waters Corporation, MA, USA) with an Agilent-C18 column (250 × 4.6 mm, 5 μm, CA, USA). A HBGK instrument equipped with a NU-3000 detector and a NP-7000C delivery system using an YMC-Pack ODS-A C18 column (250 mm × 10 mm, 5 μm) and a COSMOSIL Cholester (150 mm × 10 mm, 5 μm) was used for the semi-preparative HPLC. The TLC was performed using GF254 plates and HF-254 plates (Qingdao Marine Chemical Inc., Qingdao, China).

### 3.3. Extraction and Isolation 

Dried parts of *P. notoginseng* (1.35 kg) leaves were dissolved with EtOH, then filtered and concentrated to a certain concentration to get a sample solution, which was subjected to a silica gel column with dichloromethane-methanol-water to afford Fr.1-Fr.14. Fr.5 (40.0 g) was separated by a silica gel column with the elution of dichloromethane-methanol-water, providing the subfractions Fr.5.1-Fr.5.5. Fr.5.3 (8.2 g) was subjected to a ODS column with methanol-water to afford Fr.5.3.1-Fr.5.3.5. Fr.5.3.2 was further purified by semi-preparative HPLC (YMC ODS-A, 29% MeCN-H_2_O, v = 10.0 mL·min^−1^) to attain**18** (100.9 mg) and **19** (9.5 mg). Fr.5.3.3 (200.5 mg) was subjected to an ODS column with methanol-water (25–35%) to afford Fr.5.3.1-Fr.5.3.5. Fr.5.3.3 was also purified by semi-preparative HPLC (COSMOSIL Cholester, 31% MeCN-H_2_O, v = 2.0 mL·min^−1^) to attain **10** (14.0 mg), **11** (11.6 mg), **12** (10.0 mg), **13** (12.0 mg), and **14** (43.0 mg). Fr.5.3.5 was purified by the same method to attain **1** (4.0 mg), **2** (24.0 mg), **3** (40.5 mg), **4** (3.0 mg), and **15** (128.2 mg).

Fr.7 (42.0 g) was applied on the column of ODS, eluting with 20–100% MeOH-H_2_O, successively, providing Fr.7.1-Fr.7.6. The Fr.7.3 fraction was further separated by a silica gel column, eluting with dichloromethane-methanol-water in a gradient, and gave subfractions Fr.7.3.1-Fr.7.3.9. All parts were further separated by semi-preparative HPLC (COSMOSIL Cholester, MeCN-H_2_O, v = 2.0 mL·min^−1^) to attain **16** (34.2 mg), **17** (400.3 mg), **20** (43.2 mg), and **21** (7.0 mg). Fr.12 (12.8 g) was fractionated by ODS column (MeOH-H_2_O, 20–100%) to yield seven subfractions (Fr.12.1-Fr.12.7) with similar methods. Fr.12.2 was subjected to silica gel (CHCl_3_-MeOH-H_2_O) to provide Fr.12.2.1-12.2.5. All parts were further separated by ODS column and semipreparative HPLC (COSMOSIL Cholester, MeCN-H_2_O, v = 2.0 mL·min^−1^) to yield **5** (57.1 mg), **6** (10.0 mg), **7** (24.6 mg, **8** (6.9 mg), **9** (10.3 mg), and **22** (17.2 mg).

#### 3.3.1. Notoginsenoside SL_1_ (**1**)

White amorphous powder; ${\left[\alpha \right]}_{D}^{23}$ + 7.00 (c 0.19, MeOH); UV (MeOH) λ_max_ 192 (log ε) (5.56) nm; IR (KBr) υ_max_ 3416, 2878, 1388, 1078, 1038, and 579 cm^–1^; ^1^H and ^13^C-NMR data see Appendix A (Appendix A); (+)-HRESIMS *m*/*z* 957.5231 [M + Na]^+^ (calcd. for C_47_H_82_O_18_Na, 957.5542).

#### 3.3.2. Notoginsenoside SL_2_ (**2**)

White amorphous powder; ${\left[\alpha \right]}_{D}^{23}$ −1. 46 (c 0.20, MeOH); UV (MeOH) λ_max_ (log ε) 191 (3.95) nm; IR (KBr) υ_max_ 3406, 2878, 1645, 1265, 1230, 902, 847, 809 cm^−1^; ^1^H and ^13^C-NMR data see Appendix A; (+)-HR-ESI-MS *m*/*z* 955.5245 [M + Na]^+^ (calcd. for C_47_H_80_O_18_Na, 955.5232). 

#### 3.3.3. Notoginsenoside SL_3_ (**3**)

White amorphous powder; ${\left[\alpha \right]}_{D}^{23}$ + 7.89 (c 0.18, MeOH); UV (MeOH) λ_max_ (log ε) 217 (3.65) nm; IR (KBr) υ_max_ 3421, 2879, 1630, 1261, 895 cm^−1^; ^1^H and ^13^C-NMR data see Appendix A; (+)-HR-ESI-MS m/z 971.5255 [M + Na]^+^ (calcd. for C_47_H_80_O_19_Na, 971.5242).

#### 3.3.4. Notoginsenoside SL_4_ (**4**)

White amorphous powder; ${\left[\alpha \right]}_{D}^{23}$ + 19.20 (c 0.15, MeOH); UV (MeOH) λ_max_ (log ε) 196 (3.25) nm; IR (KBr) υ_max_ 3433, 2876, 1634, 1260, 895 cm^−1^, ^1^H and ^13^C-NMR data see Appendix A; (+)-HR-ESI-MS m/z 955.5247 [M + Na]^+^ (calcd. for C_47_H_80_O_18_Na, 955.5647). 

#### 3.3.5. Notoginsenoside SL_5_ (**5**)

White amorphous powder; ${\left[\alpha \right]}_{D}^{23}$ + 10.67 (c 0.15, MeOH); UV (MeOH) λ_max_ (log ε) 218 (3.09); IR (KBr) υ_max_ 3417, 2926, 1635, 1453, 1312, 1258, 921, 846 cm^−1^; ^1^H and ^13^C-NMR data seen Appendix A; (+)-HR-ESI-MS m/z 1265.6144 [M + Na]^+^ (calcd. for C_58_H_98_O_28_Na, 1265.6142).

#### 3.3.6. Notoginsenoside SL_6_ (**6**)

White amorphous powder; ${\left[\alpha \right]}_{D}^{23}$ + 7.5 (c 0.15, MeOH); UV (MeOH) λ_max_ (log ε) 218 (3.77); IR (KBr) υ_max_ 3415, 2879, 1636, 1454, 1313, 1258, 1199, 1158, 922, 845 cm^−1^; ^1^H and ^13^C-NMR data see Appendix A; (+)-HR-ESI-MS *m*/*z* 1265.6149 [M + Na]^+^ (calcd. for C_58_H_98_O_28_Na, 1265.6142).

#### 3.3.7. Notoginsenoside SL_7_ (**7**)

White amorphous powder; ${\left[\alpha \right]}_{D}^{24}$ + 4.40 (c 0.18, MeOH); UV (MeOH) λ_max_ (log ε) 201 (4.44) nm; IR (KBr) υ_max_ 3425, 2973, 1631, 1451, 1381, 1313, 1257, 881, 847 cm^−1^; ^1^H and ^13^C-NMR data see Appendix A; (+)-HR-ESI-MS m/z 1265.6146 [M + Na]^+^ (calcd. for C_58_H_98_O_28_Na, 1265.6142).

#### 3.3.8. Notoginsenoside SL_8_ (**8**)

White amorphous powder; ${\left[\alpha \right]}_{D}^{24}$ −7.20 (c 0.32, MeOH); UV (MeOH) λ_max_ (log ε) 203 (5.85) nm; IR (KBr) υ_max_ 3411, 2940, 1631, 1454, 1385, 1312, 1265, 894, 841 cm^−1^; ^1^H and ^13^C-NMR data see Appendix A; (+)-HR-ESI-MS m/z 1232.4306 [M + Na]^+^ (calcd. for C_58_H_96_O_26_Na, 1232.3532).

### 3.4. Hydrolysis of Sugar and Determination of Absolute Configuration

#### 3.4.1. Acid Hydrolysis

Each compound (2.0 mg each) was dissolved in 2% HCl-dioxane (1:1) for a total of 4 mL solvent, and 80 °C under the condition responded for 5 h. After the reaction, the reactants were extracted with chloroform for 3 times (3 × 2 mL). Then, the water layer was neutralized with Amberlite IRA-401 and finally filtered and vacuum concentrated to obtain monosaccharide mixture.

#### 3.4.2. Determination of Absolute Configuration

Monosaccharide mixture made the solvent in 2 mL pyridine and added L-Cysteine methyl ester hydrochloride (1.5 mg), then reacted at 60 °C for 1 h. Then, 1.5 mL N-(Trimethylsilyl) imidazole was added under the condition of an immediate ice-, and reacted at 60 °C for 30 min to obtain derivates of monosaccharide. Next, monosaccharide derivatives were prepared by the same method. The derivative of monosaccharide and the standard were analyzed by GC. By comparing the retention time of monosaccharide derivatives of samples and standards, the types and absolute configuration of sugar in samples were determined. The retention time of D/L-glucose, D/L-xylose, and D/L-arabinose is 19.817 min/21.223 min, 14.285 min/15.943 min, and 15.328 min/14.686 min.

### 3.5. Neuroprotective Effect

Glutamate (Glu) is the main transmitter of excitatory synapses in the central nervous system (CNS), and it plays an important role in excitatory of the dielectric synapse and flexibility of the synapse. Besides, it is pivotal in facilitating calcium transportation and the growth, differentiation, and recondition of intracerebral neurons [30]. Under normal circumstances, the release, ingestion, and reabsorption of glutamate are in dynamic equilibrium, whereas when glutamate is released or malabsorption occurs excessively, glutamate accumulates in the brain and causes the concentration ascension, resulting in a series of pathological reactions, finally leading to degeneration and necrosis of nerve cells [31]. Studies have confirmed that neurodegenerative diseases, such as Alzheimer disease (AD) and Parkinson disease (PD) are closely related to the neurotoxicity of glutamate [32]. Furthermore, it has been proposed that valproic acid (VPA), which is used in epileptic and bipolar disorders, may be protective against excitotoxicity insult [33]. The aerial part of *P. notoginseng* mainly contains protopanaxadiol saponins that mainly present the effect of central inhibition. Protopanaxadiol saponins are beneficial to tranquilizing, allaying excitement, and eliminating inflammation and analgesic effect. The protective effect of *P. notoginseng* leaves extract on glutamate induced SH-SY5Y cell injury was investigated in this paper.

#### 3.5.1. Cell Culture 

The study was performed using SH-SY5Y human neuroblastoma cells that were grown in Dulbecco’s modified Eagle’s medium with F12 (1:1) containing 10% fetal bovine serum (Dalian Meilun Biotech Co., Ltd., Dalian, China) and 0.1% penicillin-streptomycin solution. The cells were incubated at 37 °C with 5% CO_2_. The SH-SY5Y cells were seeded into 96-well plates (1 × 10^4^ cells/well) and incubated in complete culture medium for 24 h prior to the addition of L-glutamate or VPA.

#### 3.5.2. Treatment of Cell Damage by Glutamate 

Cells were treated with 9 different concentrations of L-glutamate (8, 9, 10, 11, 12, 13, 14, 15, and 16 µM; Shanghai Titan Scientific Co., Ltd., Shanghai, China) to determine the glutamate toxicity in the cultured SH-SY5Y cells. The glutamate concentrations that caused a significant reduction in cell viability were determined by drawing dose-cell viability curves. The glutamate concentration of 13 µM caused a ~30% decrease in cell viability after 24 h, then used this for subsequent experiment. Cell viability was determined by MTT assays, as described below. 

#### 3.5.3. Screening of VPA Concentration

SH-SY5Y cells were treated with 6 different concentrations (3.125, 6.25, 12.5, 25, 50 and 100 µM) of VPA (Depakin, 400 mg/4 mL; lyophilized powder; Sanofi S.A., Paris, Britain) for 2 h prior to exposure to 13 µM glutamate. The effect of VPA treatment also was tested by MTT assays. The VPA concentrations caused a significant increase in cell viability determined by MTT assays. Finally, the VPA concentration of 13 µM was used as the optimum concentration.

#### 3.5.4. Cell Viability Assay

The logarithmic phase cells were rinsed in phosphate-buffered saline (PBS) and digested with trypsin to make a single cell suspension. Then, the 96 well plate was made into a planked cell suspension and the cell concentration per well was 1 × 10^4^. Next, all models were randomly divided into three groups: the control group, model group (L-glutamate treated), positive control group (VPA treated), *P. notoginseng* leaves extract treated group, and 30 μM *P. notoginseng* leaves extract was added to administration group. Finally, equivalent DEME culture medium was added to the control group and model group. Six multiple wells were set in each group and all cells were incubated for 15~20 h with 5% CO_2_ at 37 °C. Then, the cell viability was evaluated by MTT assays.

#### 3.5.5. MTT Assay

An MTT (Sigma-Aldrich; Merck KGaA) assay was applied to evaluate cell viability. After adding MTT solution (5 mg/mL) to each well, cells were incubated for 4 h with 5% CO_2_ at 37 °C. Then, following the removal of the culture medium, 200 µL dimethyl sulfoxide was used to dissolve the formazan product. Finally, absorbance values were measured at 490 nm using a microplate reader (Tecan Trading Co., Ltd., Männedorf, Switzerland). Cell viability was calculated by considering the controls as 100%.

## 4. Conclusions

In conclusion, for the consideration of rational utilization of *P. notoginseng* resources, the chemical compositions of *P. notoginseng* leaves were studied mainly, especially rare saponins with altered side chains. Furthermore, their neuroprotective activities were further investigated. According to the results of the study, 22 saponins were isolated and purified, including 8 new compounds, which were all rare saponins with altered side chains. The neuroprotective activities of these saponins were determined by establishing a model of L-glutamate-induced nerve cell injury. The new compounds **1** and **3,** and known compounds **9** and **10** showed slight functions on neuroprotection. Our results not only increase the molecular diversity and bioactive diversity, but also provide a theoretical basis for promoting the rational utilization of *P. notoginseng* leaves. In addition, the study of *P. notoginseng* leaves can also promote the sustainable utilization of *P. notoginseng* medicinal resources and the economic development of the western region of China.

## Figures and Tables

**Figure 1 molecules-28-02194-f001:**
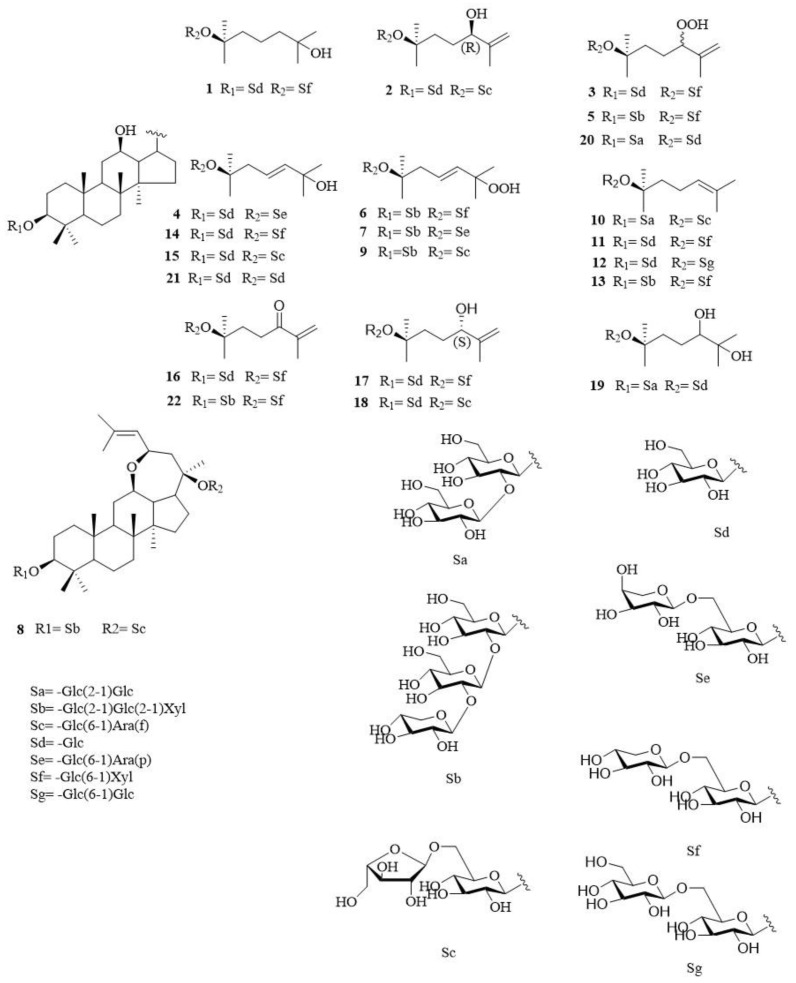
The structures of **1**–**22**.

**Figure 2 molecules-28-02194-f002:**
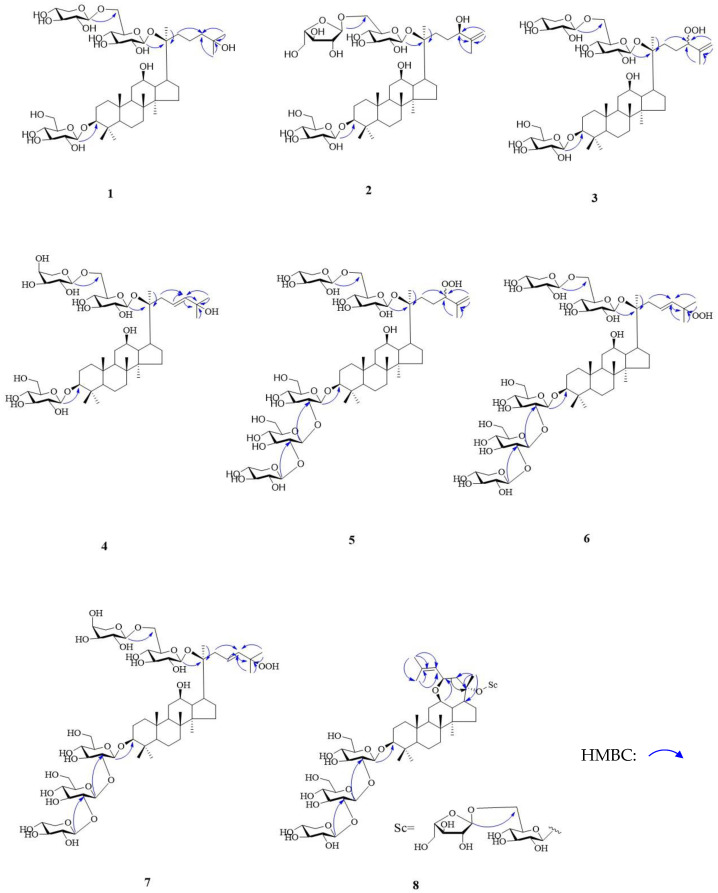
The key HMBC correlations of compounds **1**–**8**.

**Figure 3 molecules-28-02194-f003:**
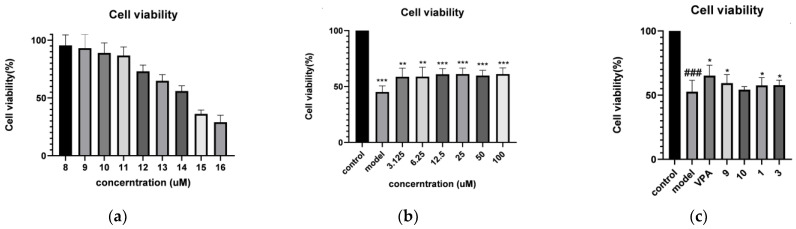
(**a**) Effect of L-glutamate pre-treatment (8, 9, 10, 11, 12, 13, 14, 15, 16 µM) on the viability of SH-SY5Y cells with glutamate-induced excitotoxicity. (**b**) Viability of SH-SY5Y cells pre-treated with VPA (3.125, 6.25, 12.5, 25, 50 µM) followed by exposure to 13 µM glutamate for 24 h. (**c**) Neuroprotective effects of compounds **1**, **3**, **9**, **10** at concentration at 30 µM in SH-SY5Y cells, respectively (** *p* < 0.01 and *** *p* < 0.001 vs. model group in **b**; ^###^ *p* < 0.001 vs. control group, * *p* < 0.05 vs. model group in **c**).

**Table 1 molecules-28-02194-t001:** ^1^H-NMR (600 MHz) and ^13^C-NMR (150 MHz) data of compound **1** (*δ* in ppm).

NO.	*δ* _C_ ^b^	*δ*_H_^b^ (*J* in Hz)	NO.	*δ* _C_ ^b^	*δ*_H_ ^b^(*J* in Hz)
1	38.9	0.73 m, 1.54 m	3-O-Glc		
2	26.5	1.80 m, 2.20 m	1′	106.8	4.97 (d, *J* = 7.8 Hz)
3	88.6	3.43 (dd, *J* = 4.8, 10.8 Hz)	2′	75.6	4.08 m
4	39.4	/	3′	78.7	4.19 m
5	56.1	0.70 (d, *J* = 11.4 Hz)	4′	71.6	4.24 m
6	18.2	1.34 m, 1.46 m	5′	78.2	4.04 m
7	34.8	1.46 m ^a^, 1.18 (d, *J* = 9.6 Hz)	6′	62.8	4.44 (dd, *J* = 6.0, 12.0 Hz)
8	39.7	/	20-O-Glc		
9	49.8	1.36 m	1″	97.9	5.12 (d, *J* = 7.8 Hz)
10	37.3	/	2″	74.8	3.93 m
11	30.7	1.51 m ^a^, 1.97 m ^a^	3″	78.5	4.25 m
12	70.2	3.65 m ^a^	4″	71.3	4.20 m
13	49.4	2.00 m	5″	76.6	4.14 m
14	51.2	/	6″	69.4	4.31 m, 4.71 (d, *J* = 11.4 Hz)
15	30.3	0.95 m, 1.83 m	Xyl		
16	26.1	1.30 m, 1.51 m	1‴	105.5	4.97 (d, *J* = 7.2 Hz)
17	51.6	1.42 m	2‴	74.7	4.02 m ^a^
18	15.8	0.90 s	3‴	77.8	4.13 m ^a^
19	16.1	0.81 s	4‴	70.9	4.14 m ^a^
20	83.0	/	5‴	66.8	3.96 (t, *J* = 10.2 Hz), 4.30 m
21	23.6	1.61 s			
22	39.8	2.80 (dd, *J* = 7.8, 13.8 Hz)			
23	26.4	1.93 m, 2.45 m			
24	27.5	1.81 m, 2.05 m			
25	70.5	/			
26	24.8	1.59 s			
27	25.2	1.53 s			
28	25.9	1.30 s			
29	16.6	0.99 s			
30	17.0	1.00 s			

^a^: Overlapped signals, ^b^: C_5_D_5_N, s: singlet, d: doublet, t: triplet, m: multiplet, ′: the first sugar, ″: the second sugar, ‴: the third sugar, /: no hydrogen.

**Table 2 molecules-28-02194-t002:** ^1^H-NMR (600 MHz) and ^13^C-NMR (150 MHz) data of compound **2** (*δ* in ppm).

NO.	*δ* _C_ ^b^	*δ*_H_^b^ (*J* in Hz)	NO.	*δ* _C_ ^b^	*δ*_H_^b^ (*J* in Hz)
1	38.9	0.76 m, 1.52 m	3-O-Glc		
2	26.5	1.82 m, 2.28 m	1′	106.8	4.97 (d, *J* = 7.8 Hz)
3	88.6	3.37 (dd, *J* = 4.8, 10.8 Hz)	2′	76.0	4.08 m ^a^
4	39.4	/	3′	78.9	4.21 m ^a^
5	56.1	0.72 m	4′	71.8	3.99 m
6	18.2	1.36 m, 1.47 m	5′	78.2	4.05 m
7	34.8	1.20 m, 1.48 m	6′	62.8	4.44 (dd, J = 4.8, 10.8 Hz)
8	39.7	/	20-O-Glc		
9	49.9	1.37 m	1″	97.9	5.17 (d, *J* = 7.7 Hz)
10	36.7	/	2″	75.1	3.97 m
11	30.1	1.40 m, 1.99 m ^a^	3″	78.8	4.30 m
12	70.1	3.12 m	4″	71.9	4.22 m
13	49.2	2.05 m	5″	76.4	4.04 m
14	51.2	/	6″	68.3	4.09 m, 4.36 m
15	30.5	0.97 m, 1.45 m	Ara(f)		
16	26.4	1.38 m, 2.24 m	1‴	109.8	4.69 br.s
17	51.7	2.54 m	2‴	83.4	4.13 m ^a^
18	15.6	0.93 s	3‴	78.7	4.16 m ^a^
19	16.0	0.79 s	4‴	85.9	4.06 m ^a^
20	83.2	/	5‴	62.7	3.71 (d, *J* = 10.2 Hz), 4.30 m
21	22.3	1.63 s			
22	32.6	2.54 m, 2.24 (dd, *J* = 9.8, 15.2 Hz)			
23	30.6	1.53 m, 2.23 m			
24	76.6	4.47 m			
25	150.0	/			
26	110.3	4.95 br.s, 5.26 br.s			
27	17.8	1.91 s			
28	27.9	1.31 s			
29	16.6	0.96 s			
30	17.0	0.94 s			

^a^: Overlapped signals, ^b^: C_5_D_5_N, s: singlet, d: doublet, m: multiplet, br.s: broad singlet, ′: the first sugar, ″: the second sugar, ‴: the third sugar, /: no hydrogen.

**Table 3 molecules-28-02194-t003:** ^1^H-NMR (600 MHz) and ^13^C-NMR (150 MHz) data of compound **3** (*δ* in ppm).

NO.	*δ* _C_ ^b^	*δ*_H_ ^b^(*J* in Hz)	NO.	*δ* _C_ ^b^	*δ*_H_^b^ (*J* in Hz)
1	38.9	0.76 m, 1.52 m	3-O-Glc		
2	26.5	1.82 m, 2.28 m	1′	106.8	4.97 (d, *J* = 7.6 Hz)
3	88.6	3.38 (dd, *J* = 4.2, 11.6 Hz)	2′	75.6	4.08 m ^a^
4	39.4	/	3′	78.7	4.19 m ^a^
5	56.1	0.73 (d, *J* = 12.4 Hz)	4′	71.6	4.24 m
6	18.2	1.36 m, 1.47 m	5′	78.2	4.04 m
7	34.8	1.18 m	6′	62.8	4.44 m, 4.64 (d, *J* = 12.4 Hz)
8	39.8	/	20-O-Glc		
9	50.0	1.37 m	1″	97.9	5.12 br.s
10	37.3	/	2″	75.1	3.93 m
11	30.7	1.51 m ^a^, 1.99 m ^a^	3″	78.8	4.25 m
12	66.9	4.18 m ^a^	4″	71.9	4.20 m ^a^
13	49.3	2.02 m	5″	76.4	4.14 m ^a^
14	51.2	/	6″	68.3	4.10 m, 4.31 m
15	30.3	0.97 m ^a^, 1.45 m ^a^	Xyl		
16	26.1	1.30 m ^a^, 2.25 m ^a^	1‴	105.5	4.97 (d, *J* = 7.2 Hz)
17	51.6	1.42 m	2‴	74.7	4.02 m
18	15.8	0.97 s	3‴	77.8	4.13 m
19	16.1	0.81 s	4‴	70.9	4.14 m
20	83.0	/	5‴	66.8	3.96 (t, *J* = 10.2 Hz), 4.30 m
21	23.6	1.63 s			
22	22.2	1.64 m ^a^			
23	26.0	1.97 m ^a^, 2.23 m ^a^			
24	89.8	4.80 m			
25	145.9	/			
26	113.3	5.09 br.s, 5.27 br.s			
27	17.3	1.96 s			
28	27.9	1.32 s			
29	16.6	1.01 s			
30	17.0	0.98 s			

^a^: Overlapped signals, ^b^: C_5_D_5_N, s: singlet, d: doublet, t: triplet, m: multiplet, br.s: broad singlet, ′: the first sugar, ″: the second sugar, ‴: the third sugar, /: no hydrogen.

**Table 4 molecules-28-02194-t004:** ^1^H-NMR (600 MHz) and ^13^C-NMR (150 MHz) data of compound **4** (*δ* in ppm).

NO.	*δ* _C_ ^b^	*δ*_H_ ^b^(*J* in Hz)	NO.	*δ* _C_ ^b^	*δ*_H_ ^b^(*J* in Hz)
1	39.1	0.73 m, 1.52 m	3-O-Glc		
2	26.8	1.34 m, 1.98 m	1′	106.8	4.93 (d, *J* = 7.8 Hz)
3	89.2	3.43 (dd, *J* = 4.8, 10.8 Hz)	2′	75.6	4.08 m ^a^
4	39.7	/	3′	78.7	4.21 m ^a^
5	56.4	0.66 m	4′	71.6	3.99 m
6	18.5	1.34 m, 1.47 m	5′	78.2	4.05 m
7	35.1	1.19 m ^a^	6′	62.8	4.44 m
8	40.1	/	20-O-Glc		
9	50.1	1.35 m	1″	97.9	5.14 (d, *J* = 7.8 Hz)
10	36.9	/	2″	75.1	3.93 m
11	30.8	1.54 m ^a^, 1.99 m ^a^	3″	78.8	4.35 m
12	70.6	3.65 m ^a^	4″	71.9	3.96 m
13	49.5	4.94 m	5″	76.4	4.04 m
14	51.5	/	6″	68.3	4.15 m, 4.25 m
15	30.5	0.97 m ^a^, 1.45 m ^a^	Ara (p)		
16	26.4	1.38 m ^a^, 2.25 m ^a^	1‴	104.6	5.00 (d, *J* = 6.0 Hz)
17	52.4	3.16 m	2‴	72.1	4.46 m
18	16.6	0.87 s	3‴	74.1	4.22 m
19	16.3	0.95 s	4‴	68.5	4.37 m
20	83.8	/	5‴	65.6	3.79 m, 4.30 m
21	23.3	1.48 s			
22	39.6	2.24 (dd, *J* = 16.0, 9.6 Hz)			
23	122.7	6.23 (ddd, *J* = 5.8, 8.5, 15.5 Hz)			
24	142.2	6.09 (d, *J* = 15.5 Hz)			
25	70.0	/			
26	17.7	1.59 s			
27	17.8	1.53 s			
28	27.9	1.30 s			
29	28.1	0.99 s			
30	17.2	1.02 s			

^a^: Overlapped signals, ^b^: C_5_D_5_N, s: singlet, d: doublet, m: multiple, ′: the first sugar, ″: the second sugar, ‴: the third sugar, /: no hydrogen.

**Table 5 molecules-28-02194-t005:** ^1^H-NMR (600 MHz) and ^13^C-NMR (150 MHz) data of compound **5** (*δ* in ppm).

NO.	*δ* _C_ ^b^	*δ*_H_ ^b^(*J* in Hz)	NO.	*δ* _C_ ^b^	*δ*_H_ ^b^(*J* in Hz)
1	39.0	0.74 m, 1.52 m	3-O-Glc		
2	26.6	1.34 m, 1.82 m	1′	104.6	4.95 (d, *J* = 7.7 Hz)
3	88.6	3.30 (dd, *J* = 3.9, 11.5 Hz)	2′	82.3	4.13 m ^a^
4	39.5	/	3′	78.5	4.37 m ^a^
5	56.1	0.68 (d, *J* = 11.9 Hz)	4′	71.4	4.11 m ^a^
6	18.2	1.34 m, 1.47 m	5′	78.1	3.99 m ^a^
7	34.9	1.20 (d, *J* = 11.9 Hz)	6′	62.7	4.39 m ^a^
8	39.8	/	-O-Glc		
9	50.0	1.35 m	1″	103.0	5.54 (d, *J* = 6.8 Hz)
10	36.7	/	2″	84.2	4.23 m ^a^
11	30.7	1.51 m ^a^, 1.89 m ^a^	3″	77.7	4.31 m ^a^
12	69.9	4.18 m ^a^	4″	71.1	4.24 m ^a^
13	49.3	2.00 m	5″	77.6	4.16 m ^a^
14	51.1	/	6″	62.5	4.03 m, 4.39 m ^a^
15	30.4	0.96 m ^a^, 1.42 m ^a^	-Xyl		
16	26.4	1.34 m ^a^, 2.30 m ^a^	1‴	106.2	5.44 (d, *J* = 6.8 Hz)
17	51.2	2.57 m	2‴	75.8	4.13 m ^a^
18	15.8	0.96 s	3‴	78.5	4.37 m ^a^
19	16.1	0.80 s	4‴	70.9	4.11 m ^a^
20	83.2	/	5′′′	67.2	3.71 (d, *J* = 10.2 Hz)
21	22.1	1.64 s	20-O-Glc		
22	32.7	1.90 m, 2.58 m	1′′′′	97.8	5.12 br.s
23	26.4	1.97 m ^a^, 2.20 m ^a^	2′′′′	74.7	3.88 m ^a^
24	90.0	4.80 (t, *J* = 6.7 Hz)	3′′′′	78.1	4.17 m ^a^
25	146.1	/	4′′′′	71.5	4.24 m ^a^
26	113.1	5.09 br.s, 5.28 br.s	5′′′′	76.6	3.99 m ^a^
27	17.3	1.96 s	6′′′′	69.9	4.30 m ^a^, 4.76 (d, *J* = 10.0 Hz)
28	27.8	1.28 s	-Xyl		
29	16.4	1.11 s	1′′′′′	105.5	5.01 (d, *J* = 7.4 Hz)
30	17.2	0.95 s	2′′′′′	74.6	4.06 m ^a^
			3′′′′′	78.4	4.37 m ^a^
			4′′′′′	70.5	4.17 m ^a^
			5′′′′′	66.8	3.76 (t, *J* = 10.2 Hz), 4.33 m ^a^

^a^: Overlapped signals, ^b^: C_5_D_5_N, s: singlet, d: doublet, t: triplet, m: multiplet, br.s: broad singlet, ′: the first sugar, ″: the second sugar, ‴: the third sugar, ′′′′: the fourth sugar, ′′′′′: the fifth sugar, /: no hydrogen.

**Table 6 molecules-28-02194-t006:** ^1^H-NMR (600 MHz) and ^13^C-NMR (150 MHz) data of compound **6** (*δ* in ppm).

NO.	*δ* _C_ ^b^	*δ*_H_ ^b^(*J* in Hz)	NO.	*δ* _C_ ^b^	*δ*_H_^b^ (*J* in Hz)
1	38.9	0.74 m, 1.52 m	3-O-Glc		
2	26.5	1.33 m, 1.81 m	1′	104.6	4.96 (d, *J* = 6.2 Hz)
3	88.6	3.30 (dd-like)	2′	82.7	4.15 m ^a^
4	39.5	/	3′	77.7	4.39 m ^a^
5	56.1	0.69 (d, *J* = 11.7 Hz)	4′	71.4	4.14 m ^a^
6	18.2	1.38 m, 1.54 m ^a^	5′	77.9	3.99 m ^a^
7	34.8	1.21 (d, *J* = 11.0 Hz)	6′	62.7	4.51 m ^a^
8	39.8	/	-O-Glc		
9	49.9	1.38 (d, *J* = 11.7 Hz)	1″	102.9	5.54 (dd-like)
10	36.7	/	2″	84.3	4.24 m ^a^
11	30.7	1.54 m ^a^, 1.90 m ^a^	3″	78.1	4.31 m ^a^
12	70.2	4.07 m ^a^	4″	70.9	4.24 m ^a^
13	49.4	2.02 m ^a^	5″	77.6	4.16 m ^a^
14	51.2	/	6″	62.5	4.03 m, 4.39 m ^a^
15	30.3	0.98 m ^a^, 1.44 m ^a^	-Xyl		
16	26.1	1.46 m ^a^, 2.30 m ^a^	1‴	106.2	5.45 (d, *J* = 6.0 Hz)
17	51.6	2.47 m	2‴	75.8	4.14 m ^a^
18	15.8	1.01 s	3‴	78.5	4.39 m ^a^
19	16.1	0.83 s	4‴	71.3	4.14 m ^a^
20	83.0	/	5‴	67.2	3.72 m ^a^
21	23.0	1.63 s	20-O-Glc		
22	39.8	2.85 m, 3.13 m ^a^	1′′′′	98.0	5.20 br.s
23	126.5	6.20 m ^a^	2′′′′	74.8	3.95 m ^a^
24	137.8	6.15 br.s	3′′′′	78.1	4.21 m ^a^
25	81.1	/	4′′′′	71.5	4.24 m ^a^
26	24.8	1.62 s	5′′′′	76.6	3.99 m ^a^
27	25.2	1.62 s	6′′′′	69.7	4.35 m ^a^ 4.76 (d, *J* = 11.2 Hz)
28	27.8	1.29 s	-Xyl		
29	16.3	1.12 s	1′′′′′	105.4	5.00 (d, *J* = 6.8 Hz)
30	17.0	0.92 s	2′′′′′	74.7	4.05 m ^a^
			3′′′′′	78.8	4.39 m ^a^
			4′′′′′	70.5	4.17 m ^a^
			5′′′′′	66.8	3.72 m ^a^, 4.34 m ^a^

^a^: Overlapped signals, ^b^: C_5_D_5_N, s: singlet, d: doublet, m: multiplet, br.s: broad singlet, ′: the first sugar, ″: the second sugar, ‴: the third sugar, ′′′′: the fourth sugar, ′′′′′: the fifth sugar, /: no hydrogen.

**Table 7 molecules-28-02194-t007:** ^1^H-NMR (600 MHz) and ^13^C-NMR (150 MHz) data of compound **7** (*δ* in ppm).

NO.	*δ* _C_ ^b^	*δ*_H_^b^ (*J* in Hz)	NO.	*δ* _C_ ^b^	*δ*_H_^b^ (*J* in Hz)
1	38.9	0.73 m, 1.52 m	3-O-Glc		
2	26.6	1.40 m, 1.95 m	1′	104.6	4.95 (d, *J* = 6.2 Hz)
3	88.6	3.30 (dd, *J* = 4.2, 11.4 Hz)	2′	82.7	4.13 m ^a^
4	39.5	/	3′	77.6	4.20 m ^a^
5	56.1	0.68 (d, *J* = 11.3 Hz)	4′	71.4	4.14 m
6	18.2	1.37 m, 1.55 m ^a^	5′	77.6	3.90 m
7	34.9	1.20 (d, *J* = 11.0 Hz)	6′	62.7	4.39 m ^a^, 4.60 m ^a^
8	39.8	/	-O-Glc		
9	49.9	1.38 (d, *J* = 11.7 Hz)	1″	102.9	5.52 (d, *J* = 7.8 Hz)
10	36.7	/	2″	84.2	4.24 m ^a^
11	30.7	1.53 m ^a^, 1.92 m ^a^	3″	78.1	3.95 m ^a^
12	70.5	4.13 m ^a^	4″	70.9	4.07 m ^a^
13	49.4	2.00 m	5″	77.6	4.26 m ^a^
14	51.2	/	6″	62.5	4.00 m, 4.37 m ^a^
15	30.6	0.97 m ^a^, 1.42 m ^a^	-Xyl		
16	26.4	1.80 m ^a^, 2.47 m ^a^	1‴	106.2	5.43 (d, *J* = 6.6 Hz)
17	51.6	2.57 m ^a^	2‴	75.8	4.12 m ^a^
18	15.8	0.79 s	3‴	78.5	4.39 m ^a^
19	16.1	0.93 s	4‴	71.3	4.12 m ^a^
20	83.3	/	5‴	67.2	3.79 m ^a^
21	22.4	1.63 s	-20-O-Glc		
22	39.6	1.85 m	1′′′′	98.0	5.20 br.s
23	126.2	5.20 m ^a^	2′′′′	74.8	3.95 m ^a^
24	137.8	6.15 br.s	3′′′′	78.5	4.33 m ^a^
25	81.1	/	4′′′′	71.3	3.92 m ^a^
26	24.8	1.61 s	5′′′′	76.4	3.99 m ^a^
27	25.2	1.61 s	6′′′′	68.7	4.25 m ^a^ 4.76 (d, *J* = 11.2 Hz)
28	27.8	1.27 s	-Ara (p)		
29	16.3	1.14 s	1′′′′′	104.1	5.00 (d, *J* = 6.8 Hz)
30	17.0	0.94 s	2′′′′′	71.8	4.55 m ^a^
			3′′′′′	73.9	4.23 m ^a^
			4′′′′′	68.3	4.37 m ^a^
			5′′′′′	65.1	3.79 m ^a^, 4.31 m ^a^

^a^: Overlapped signals, ^b^: C_5_D_5_N, s: singlet, d: doublet, m: multiplet, br.s: broad singlet, ′: the first sugar, ″: the second sugar, ‴: the third sugar, ′′′′: the fourth sugar, ′′′′′: the fifth sugar, /: no hydrogen.

**Table 8 molecules-28-02194-t008:** ^1^H-NMR (600 MHz) and ^13^C-NMR (150 MHz) data of compound **8** (*δ* in ppm).

NO.	*δ* _C_ ^b^	*δ*_H_^b^ (*J* in Hz)	NO.	*δ* _C_ ^b^	*δ*_H_ ^b^(*J* in Hz)
1	38.9	0.86 m, 1.56 m	3-O-Glc		
2	26.7	1.76 m, 2.19 m	1′	105.3	4.96 (d, *J* = 8.3 Hz)
3	89.1	3.30 (dd, *J* = 4.2, 11.4 Hz)	2′	83.2	4.23 m ^a^
4	39.1	/	3′	77.4	4.30 m ^a^
5	56.5	0.61 (d, *J* = 11.9 Hz)	4′	71.4	4.24 m ^a^
6	18.5	1.50 m, 1.40 m ^a^	5′	78.1	3.94 m ^a^
7	35.2	1.47 m, 1.15 m	6′	62.8	4.34 m ^a^, 4.47 m ^a^
8	40.1	/	-O-Glc		
9	50.2	1.51 (d, *J* = 11.7 Hz)	1″	105.6	5.40 (d, *J* = 6.9 Hz)
10	37.0	/	2″	75.2	4.08 m ^a^
11	30.9	1.34 m ^a^, 1.80 m ^a^	3″	77.3	4.37 m ^a^
12	79.1	3.72 m ^a^	4″	71.7	4.13 m ^a^
13	49.5	1.38 m ^a^	5″	78.3	3.99 m ^a^
14	51.6	/	6″	62.6	4.22 m, 4.47 m
15	31.1	1.15 m ^a^, 1.43 m ^a^	-Xyl		
16	26.9	2.16 m ^a^, 2.24 m ^a^	1‴	106.2	5.44 (d, *J* = 6.6 Hz)
17	49.8	3.61 m ^a^	2‴	75.1	4.30 m ^a^
18	15.9	0.88 s	3‴	78.5	4.37 m ^a^
19	16.4	0.92 s	4‴	70.4	4.04 m ^a^
20	83.6	/	5‴	65.8	3.75 m ^a^
21	22.8	1.61 s	20-O-Glc		
22	52.7	2.16 m, 2.24 m	1′′′′	98.4	5.18 br.s
23	71.9	4.83 (t, *J* = 8.2 Hz)	2′′′′	74.8	3.95 m ^a^
24	129.4	6.02 (d, *J* = 7.6 Hz)	3′′′′	78.5	4.33 m ^a^
25	131.6	/	4′′′′	78.3	3.99 m ^a^
26	29.7	1.65 s	5′′′′	71.3	4.70 m ^a^
27	19.5	1.94 s	6′′′′	67.2	4.29 m ^a^ 4.76 (d, *J* = 11.2 Hz)
28	28.2	1.27 s	-Ara (f)		
29	16.7	1.10 s	1′′′′′	110.3	5.68 (d, *J* = 6.8 Hz)
30	17.4	0.79 s	2′′′′′	83.5	4.27 m ^a^
			3′′′′′	78.8	4.41 m ^a^
			4′′′′′	85.9	4.23 m ^a^
			5′′′′′	62.9	4.30 m ^a^, 4.40 m ^a^

^a^: Overlapped signals, ^b^: C_5_D_5_N, s: singlet, d: doublet, t: triplet, m: multiplet, br.s: broad singlet, ′: the first sugar, ″: the second sugar, ‴: the third sugar, ′′′′: the fourth sugar, ′′′′′: the fifth sugar, /: no hydrogen.

## Data Availability

Not applicable.

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
