# Peer review of "Study on Chemical Constituents of Panax notoginseng Leaves"

_molecules, 2023, doi:10.3390/molecules28052194_

Round 1

Reviewer 1 Report

1. Line 44; change “Therefor” to “Therefore”, Line 61; change “silic” to “silica”, Line 98; change “formular” to “formula” and also in the whole paper.

2. Line 376; change “kown” to “known”, Line 378; change “uitraviolet” to “ultraviolet”, Line 484; change “4.4Neuroprotective effect” to “4.5Neuroprotective effect”.

3. Lines 501-534; please number the following titles Cell culture, Treatment of cell damage by glutamate, Screening of VPA concentration, Cell viability assay, MTT assay.

4. Write a conclusion part for the paper

5. The cited references are little please mention more references. 

Author Response

Dear Reviewer,   

   We feel great thanks for your professional review work on our article. As you are concerned, there are several problems that need to be addressed. According to your nice suggestions, we have made extensive corrections to our previous draft, and the detailed corrections are listed below.

Point 1: Line 44; change “Therefor” to “Therefore”, Line 61; change “silic” to “silica”, Line 98; change “formular” to “formula” and also in the whole paper.

Response 1: I am very sorry for our incorrect writing, and thank you for your reminder. I have corrected “Therefor” to “Therefore” (Line 47), “silic” to “silica” (Line 66), “formular” to “formula” (Lines 74, 103, 142, 179, 212, 244, 281 and 313) and also in whole paper.

Point 2: Line 376; change “kown” to “known”, Line 378; change “uitraviolet” to “ultraviolet”, Line 484; change “4.4 Neuroprotective effect” to “4.5 Neuroprotective effect”.

Response 2: I feel sorry for my carelessness. In my resubmitted manuscript, the typo is revised. I have changed “kown” to “known” (Line 374), “uitraviolet” to “ultraviolet” (Line 376), and I have corrected the number. Thanks for your guidance.

Point 3: Lines 501-534; please number the following titles Cell culture, Treatment of cell damage by glutamate, Screening of VPA concentration, Cell viability assay, MTT assay.

Response 3: I sincerely thank you for careful reading. As suggested, I have numbered the titles.

Point 4: Write a conclusion part for the paper.

Response 4: I sincerely appreciate the valuable comments. I have checked the manuscript carefully, and added a conclusion part for the paper (Line 536).

Point 5: The cited references are little please mention more references.

Response 5: As suggested by the Reviewer 1, I have checked the literature carefully and added more references (references 1, 4, 6, 8, 10, 11, 13, 14) on Panax notoginseng leaves into the Introduction part to support this manuscript.

  We tried our best to improve the manuscript and made some changes marked in highlight in revised paper which will not influence the content and framework of the paper. We appreciate for Reviewer's warm work earnestly, and hope the corrections will meet with approval. Once again, thank you very much for your comments and suggestions.

   Best wishes.

                                                                                                      Yours sincerely, 

                                                                                                       Sun Xiaojuan 

Reviewer 2 Report

The review of the paper entitled:

“Study on chemical constituents of Panax notoginseng leaves”

The manuscript described the structural elucidation of eight new dammarane saponins namely Notoginsenosides SL1-SL8 (1-8) from the leaves of P. notoginseng together with fourteen known compounds. Biological assay showed four compounds 1, 3, 9 and 10 showed slight protective effects against L-glutamate-induced nerve cell injury (30 µM).   

The results of study are good. However, there are many English typo errors in the manuscript. The authors should check the manuscript carefully. I recommend the manuscript will be reconsidered after major revisions.

1.      Line 15: Correct “never’’ to nerve

2.      Line 16, 22: Notoginsenosides

3.      Line 23; against

4.      Line 48: mainly

5.      Line 58: neuroprotective

6.      Line 61: silica

7.      Line 63: Yielded

8.      Line 66: Structures. The authors should check the structures of compounds  1-8, especially the sugar parts Sa-Sf are not mateched with Figure 2.

9.      Line 74: field

10.  Line 84: R,S, α, β in italic. The authors sould check in the entire manuscript.

11.  Line 86, 124 and structural elucidation parts of comound 3-8 (194,: Correct “terminal’’ as anomeric

12.  Line 98,106. 185: Compound

13.  Table 2, 3 and 5: Check the peaks of H-26, there are 2 broad peaks in HSQC.

14.  Line 145: Correct “methoxy’’ as “oxymethine”

15.  Line 155-156: Hydroxyperoxy

16.  Line 173: Table 3 is located above the table

17.  Line 204: Remove β-D-glucose

18.  Line 213: revealed

19.  Line 230: Correct Two glucoses as three glucoses

20.  Check the sentences 232-237 to match with the structure of 5.

21.  Line 267: connected

22.  Line 295-296: Check the sentence after “as well as”

23.  Check the sentences 232-237 to match with the structure of 5.

24.  Line 307-308: Correct the sentence Xyl C-1’’’ correlated with Glc C-2’’

25.  Line 3776: known

26.  Line 374; Compounds

27.  Line 546: Suppported

Author Response

Dear Reviewer,

    On behalf of all the contributing authors, I would like to express our sincere appreciations of your constructive comments concerning our article entitled "Study on Chemical Constituents of Panax notoginseng Leaves" (Manuscript No: molecules-2216549). These comments are all valuable and helpful for improving our article. According to the associate editor and reviewers' comments, we have made extensive modifications to our manascript and supplemented extra data to make our results convincing.  In the revised version, changes to our manuscript were all highlighted. Point-by-point responses are listed below.

Point 1: Line 15: Correct “never’’ to nerve.

Response 1: I am really sorry for my careless mistakes. Thank you for your reminder. I have corrected “never’’ to “nerve” (Line 16).

Point 2: Line 16, 22: Notoginsenosides.

Response 2: I think this is an excellent suggestion. I have modified all the first letter of the compound name in the paper (Line 17).

Point 3: Line 23; against.

Response 3: I apologize for the poor language of my manuscript. I have corrected “aginst” to “against” (Line 23).

Point 4: Line 48: mainly.

Response 4: Thank you for your insightful comment, and I have modified the word of “mainly” (Line 51).

Point 5: Line 58: neuroprotective.

Response 5: I am very sorry for my negligence, and I have made correction according suggestion (Line 63).

Point 6: Line 61: silica.

Response 6: I sincerely thank the reviewer for careful reading. As suggested by the reviewer, I have corrected the “silic” to “silica” (Line 66).

Point 7: Line 63: Yielded.

Response 7: I’m so sorry for my mistake, and I have modified “yield” to “yielded” (Line 68).

Point 8: Line 66: Structures. The authors should check the structures of compounds 1-8, especially the sugar parts Sa-Sf are not marched with Figure 2.

Response 8: Thanks for your careful checks. I am sorry for my careless. Based on comments, I have made the corrections of Figure 2 (Line 70).

Point 9: Line 74: field.

Response 9: I a sorry for my careless, and I have corrected “filed” to “field” (Line 79).

Point 10: Line 84: R, S, α, β in italic. The authors should check in the entire manuscript.

Response 10: I am very sorry for my incorrect writing, and I have corrected all in the paper.

Point 11: Line 86, 124 and structural elucidation parts of compound 3-8 (194: Correct “terminal’’ as anomeric.

Response 11: It’s an excellent suggestion, and I have corrected all “terminal’’ as “anomeric” (Line 91).

Point 12: Line 98,106. 185: Compound.

Response 12: I am so sorry for my careless mistakes, and I have revised all in the paper (Lines 103, 111, 187).

Point 13: Table 2, 3 and 5: Check the peaks of H-26, there are 2 broad peaks in HSQC.

Response 13: I feel great thanks for professional suggestion. According to suggestion, I have re-checked the HSQC, and have corrected all in the paper (Table 2, Table 3, Table 5).

Point 14: Line 145: Correct “methoxy’’ as “oxymethine”.

Response 14: I sincerely thank the valuable feedback, and I have made correction (Line 147).

Point 15: Line 155-156: Hydroxyperoxy.

Response 15: I apologize for the poor language of my manuscript. I have checked the words and corrected “hydroperxyl’’ to “hydroxyperoxy” (Line 154).

Point 16: Line 173: Table 3 is located above the table.

Response 16: Thanks for your suggestion, and I am sure Table 3 is located above the table now.

Point 17: Line 204: Remove β-D-glucose.

Response 17: I sincerely appreciate the valuable comment. I have removed a β-D-glucose, and have corrected β-D-glucose to β-D-glucoses (Line 205).

Point 18: Line 213: revealed.

Response 18: I am sorry for my careless, and I have corrected (Line 219).

Point 19: Line 230: Correct Two glucoses as three glucoses.

Response 19: I think this is an excellent suggestion, and I have corrected “two glucoses” as “three glucoses” (Line 233).

Point 20: Check the sentences 232-237 to match with the structure of 5.

Response 20: I sincerely thank the reviewer for careful reading. In our resubmitted manuscript, I have re-written this part according to suggestion (Lines 234-237).

Point 21: Line 267: connected.

Response 21: I am so sorry for my careless, and I have corrected (Lines 268, 269).

Point 22: Line 295-296: Check the sentence after “as well as”.

Response 22: I sincerely appreciate the valuable comment, and I have re-written this sentence in the revised manuscript (Line 296).

Point 23: Check the sentences 232-237 to match with the structure of 5.

Response 23: I sincerely thank the reviewer for careful reading. In our resubmitted manuscript, I have re-written this part according to suggestion (Lines 234-237).

Point 24: Line 307-308: Correct the sentence Xyl C-1’’’ correlated with Glc C-2’’.

Response 24: Thanks for your suggestion, and I have modified this sentence (Line 304).

Point 25: Line 3776: known.

Response 25: I feel sorry for my careless. I have re-checked and corrected this word (Line 373).

Point 26: Line 374; Compounds.

Response 26: I have checked and corrected (Line 357).

Point 27: Line 546: Suppported.

Response 27: I sincerely thank the reviewer for careful reading, and I have corrected “suppoetrd’’ to “supported” (Line 561).

  We have studied reviewers' comments carefully and have made revisions which highlighted in the paper. We have tried our best to revise our manuscript according to the comments. Attached please find the revised version, which we would like to submit for your kind consideration.

  We would like to express our great appreciation to you and other reviwers for comments on our paper. 

  Thank you and best regards.

                                                                                                      Yours sincerely,

                                                                                                        Sun Xiaojuan

Round 2

Reviewer 2 Report

The authors have made correction to improve the quality of manuscript.

However, the authors should correct some remained errors.

Line 72: Please change to “structure”

Line 100: Please change to “applied”

Line 135- 136: Re-write the sentence “Finally confirmed the structure of compound 2 and namely notoginsenoside SL2.”

Line 154, 226: Please correct “Combing” as “combining”

Figure 2: Sd in this figure is not matched with figure 1. Please correct

Line 170-171: Re-write the sentence “Finally confirmed the structure of compound 3…

Line 209-210: Re-write the sentence “Finally confirmed the structure of compound 4

Line 244-245: Re-write the sentence “Finally confirmed the structure of compound 5

Line 274-275 Re-write the sentence “Finally confirmed the structure of compound 6

Line 313-314: Re-write the sentence “Finally confirmed the structure of compound 7

Line 344-345 Re-write the sentence “Finally confirmed the structure of compound 8

Line 371: Please re-write the discussion: this part looks like the conclusion.

Author Response

Dear Reviewer,

  We would like to extend our most sincere appreciation to your valuable comments. According to the comments, we have improved the quality of our work, and highlighted the changes in the revised manuscript. We have checked all references in the paper, and we are sure that all references are relevant to the contents of the manuscript. In our manuscript, the references 1-18 are relevant to the context of the study. The references 19-29 are used for the structure determination of compounds. The references 30-33 are relevant to neuroprotection. If you have any better suggestions, please don't hesitate to let me know. Please see below for a point-by-point response to the comments.

Point 1: Line 72: Please change to “structure”.

Response 1: I am very sorry for our incorrect writing, and thank you for your reminder. I have corrected “struactures” to “structures” (Line 72).

Point 2: Line 100: Please change to “applied”.

Response 2: I feel sorry for my carelessness. In my resubmitted manuscript, the typo is revised. I have changed “applyed” to “applied” (Line 100).

Point 3: Line 135- 136: Re-write the sentence “Finally confirmed the structure of compound 2 and namely notoginsenoside SL2.

Response 3: I sincerely thank you for careful reading. As suggested, I have re-written this sentence in the passive voice in my resubmitted manuscript (Line 135-136).

Point 4: Line 154, 226: Please correct “Combing” as “combining”.

Response 4: I sincerely thank the reviewer for careful reading. I have corrected in my manuscript (Line 154, 226).

Point 5: Figure 2: Sd in this figure is not matched with figure 1. Please correct.

Response 5: I’m so sorry for my mistake, and I have rechecked and corrected in my manuscript (Fig.2).

Point 6: Line 170-171: Re-write the sentence “Finally confirmed the structure of compound 3…

Response 6: It’s an excellent suggestion, and I have re-written this sentence in passive voice in my manuscript (Line 170-171).

Point 7: Line 209-210: Re-write the sentence “Finally confirmed the structure of compound 4…

Response 7: I feel great thanks for professional suggestion, and I have re-written this sentence in passive voice in my manuscript (Line 209-210).

Point 8: Line 244-245: Re-write the sentence “Finally confirmed the structure of compound 5…

Response 8: I apologize for the poor language of my manuscript. I have re-written this sentence in passive voice in my manuscript (Line 244-245).

Point 9: Line 274-275: Re-write the sentence “Finally confirmed the structure of compound 6…

Response 9: Thanks for your suggestion, and I have re-written this sentence in passive voice in my manuscript (Line 274-275).

Point 10: Line 313-314: Re-write the sentence “Finally confirmed the structure of compound 7…

Response 10: I sincerely appreciate the valuable comment, and I have re-written this sentence in passive voice in my manuscript (Line 313-314).

Point 11: Line 344-345: Re-write the sentence “Finally confirmed the structure of compound 8…

Response 11: Thanks for your suggestion, and I have re-written this sentence in passive voice in my manuscript (Line 344-345).

Point 12: Line 371: Please re-write the discussion: this part looks like the conclusion.

Response 12: I feel great thanks for professional suggestion., and I have re-written the discuss in my manuscript (Line 369-397).

  Again, thank you for giving us the opportunity to strengthen our manuscript with your valuable comments and queries. We have work hard to incorporate your feedback and hope that these revisions persuade you to accept our submission. 

  Best regards.

                                                                                                         Yours sincerely,

                                                                                                          Sun Xiaojuan
